# OpenReview forum: "Cascade Reward Sampling for Efficient Decoding-Time Alignment"
_ICLR.cc/2025/Conference — Submitted to ICLR 2025_

### Official Review · Reviewer_udDg · 2024-10-24

**Soundness:** 1
**Presentation:** 3
**Contribution:** 2
**Rating:** 6
**Confidence:** 5

**Summary:**

This paper focuses on reward-guided decoding-time alignment. They propose Cascade Reward Sampling (CARDS), which is mainly based on two claims:

- They claim to have analyzed the properties of reward models (RMs) on incomplete text, and hypothesize that RMs can serve as approximations for value functions.
- They show that values of predictive uncertainty can be a segmentation signal, and thus can divide the generation of an entire response into multiple steps (segments).

Combining two claims, they establish the target distribution for sampling a new semantic segment. Therefore, they finally sample one segment each step, get the reward, and use rejection sampling to align the distribution to what they have established.

Experiments are conducted on HH-RLHF dataset. They have compared the performance of CARDS with RAD/ARGS and naive rejection sampling, showing general advantages.

**Strengths:**

## Originiality
- The segmentation trick with uncertainty is novel and interesting, and it deserves to be shared with a broad audience.
- The assumption on reward distribution is original, though may not be true.

## Clarity
- This paper is well-written and well-organized.
- Most figures and tables are friendly to the readers.
- The intuitions are clearly expressed.

## Significance
- The weak assumption on the expressibility of reward model is more reasonable than prior works.
- This work has improved the performance of reward-guided decoding-time alignment, compared with baselines.

**Weaknesses:**

## Major
- The reviewer cannot buy the idea that reward model is a value function. Only the last signal of reward model is trained, and the intermediate prediction is more like black box. There is a well-known paper [5] showing that DPO model acts like Q-function, but the result only holds for credit assignment setting.
- There is a new paper showing that "partial reward" may not be correlated to "full reward". Please see Appendix C.3 of [6].
- There is no non-trivial mathematical proof. So it would be better to remove the claim that "We first **rigorously** analyze the properties of reward models". The authors have conducted some experimets to support their intuitions or assumptions, that are great, but still far from being called "rigorous", since there is no theoretical guarantee.
- Lack of baselines. CARDS is only compared with two baselines, RAD/ARDS (these two are the same), and naive rejection sampling. Many reward-guided decoding-time alignment approaches have emerged in these two years. For example, controlled-decoding [1] can be a baseline, which samples next token from $\pi(y)\exp(r(y)/\beta)$. ([1] is different from RAD/ARDS, since it doesn't need a top-k sampling at first.) And the reviewer guesses that [1] should be faster than CARDS.
## Minor
- Equation 11(b) only holds for soft-RL, which should be made clear to the readers.
- Lack of datasets. The experimental results on HH-RLHF are acceptable but limitted. More datasets are worth exploring. For example, Ultrafeeback[2], HelpSteer[3], Beavertails[4] are worth trying.

*The empirical approach is generally fine to the reviewer, but unfortunately, the claims have many problems. The reviewer is willing to raise the rating if the issues can be solved.*

[1] Controlled Decoding from Language Models, https://arxiv.org/abs/2310.17022

[2] UltraFeedback: Boosting Language Models with Scaled AI Feedback, https://arxiv.org/abs/2310.01377

[3] HelpSteer: Multi-attribute Helpfulness Dataset for SteerLM, https://arxiv.org/abs/2311.09528

[4] BeaverTails: Towards Improved Safety Alignment of LLM via a Human-Preference Dataset, https://arxiv.org/abs/2307.04657

[5] From r to Q∗: Your Language Model is Secretly a Q-Function, https://arxiv.org/abs/2404.12358

[6] TreeBoN: Enhancing Inference-Time Alignment with Speculative Tree-Search and Best-of-N Sampling, https://arxiv.org/abs/2410.16033

**Questions:**

- The equation (4) is not well defined. The expectation is taken based on which distribution?
- Which reward model did you use for Figure 2b,3,4? Did you only examine one reward model to support claims shown in these figures?
- Can this approach be applied to some smaller and weaker reward models, like GPT2 models [1][2]? This would be important for some groups with restricted computation resources.
- Can this approach be possibly applied to multi-objective alignment, like what many reward-guided decoding-time algorithms [3][4] can do? (Anyway, they are concurrent works, so there is no need to compete with them.)

[1] https://huggingface.co/Tristan/gpt2_reward_summarization

[2] https://huggingface.co/Ray2333/gpt2-large-helpful-reward_model

[3] PAD: Personalized Alignment at Decoding-Time, https://arxiv.org/abs/2410.04070

[4] GenARM: Reward Guided Generation with Autoregressive Reward Model for Test-time Alignment, https://arxiv.org/abs/2410.08193

**Details Of Ethics Concerns:**

No concerns, thanks.

---

> ### Author Response · Authors · 2024-11-22
>
> **Q1: The reviewer cannot buy the idea that the reward model is a value function.**
>
> We did not claim that the reward model (RM) is a value function. We claim that RM is a good approximation to the value function on semantically complete prefixes. The core insight to understand this is that semantically complete prefixes closely resemble complete sentences and can be comprehended by RMs. This is **not** a mathematical claim, but an **explanation** based on empirical findings.
>
> The mentioned [1] primarily focuses on DPO which uses an implicit RM, while our analysis mainly focuses on explicit RM. It is unclear to us how the theoretical results on DPO in [1] would rule out our explanations for explicit RMs.
>
> [1] From r to Q*: Your Language Model is Secretly a Q-Function. 2024.
>
> **Q2: There is a new paper showing that "partial reward" may not be correlated to "full reward".**
>
> We are grateful that [2] has further verified this conclusion. As demonstrated by [2], semantically complete segmentation induces a **higher** correlation coefficient than the 1/3 cutoff.
>
> Please note that [2] uses **different QA datasets** than those used in this paper. This new setting requires **re-tuning the uncertainty threshold** to get an appropriate segmentation, much like tuning hyperparameters in many ML algorithms. In practice, we find it effective to adjust the threshold so that each response is split into no more than 10 and no fewer than 5 segments. However, [2] sets the uncertainty threshold to be the same as the HH-RLHF dataset. We suspect this is the main reason the correlation rise shown in [2] is not significant enough.
>
> We understand that the analysis of RM properties is still a growing field, and many papers may have varying conclusions. However, there is no gold-standard way to analyze this problem for now. We are open to discussing any detailed setting in studying this problem, and we hope the reviewer can acknowledge the new findings and insights of our paper.
>
> [2] TreeBoN: Enhancing Inference-Time Alignment with Speculative Tree-Search and Best-of-N Sampling, Arxiv 2024.
>
> **Q3: It would be better to remove the “rigorous”.**
>
> Thanks for the suggestion! We will remove “rigorous” and emphasize that the analysis in this paper is based on empirical findings.
>
> **Q4: Lack of baselines.**
>
> Besides RAD/ARDS and naive rejection sampling, we have also included more baselines of alignment methods in Table 2, Table 3, and Table 5 (e.g., PPO [4], DPO [5], RAIN [6]). Since the particular problem studied in this paper is alignment rather than general controlled decoding, we think the current set of baselines is already comprehensive.
>
> CD in [3] requires additional training to learn $V_{\theta}$, which introduces significant computational costs. Unlike CD, CARDS and the main baselines considered in this paper do not require any training.
>
> Furthermore, in CD, sampling the next token from $\pi(y)exp(r(y)/\beta)$ requires computing the $V_{\theta}$ over all tokens in the vocabulary. For example, if the vocabulary contains 10,000 tokens, the $V_{\theta}$ needs to be computed 10,000 times parallelly to score each candidate token. The “one call” in [3] contains multiple parallel $V_{\theta}$ computations. We think top-$K$ sampling can reduce the number of $V_{\theta}$ computations to $K$, which means CD is slower than ARGS/RAD. Since CARDS is already faster than ARGS/RAD, we believe CARDS will be faster than CD [3] under the same model size.
>
>
> [3] Controlled Decoding from Language Models. ICML 2024.
>
> [4] Proximal Policy Optimization Algorithms. 2017.
>
> [5] Direct preference optimization: Your language model is secretly a reward model. NeurIPS 2024.
>
> [6] RAIN: Your Language Models Can Align Themselves without Finetuning. ICLR 2024.
>
> **Q5: Equation 11(b) only holds for soft-RL.**
>
> Thanks for pointing out this confusion! We will add a remark on the soft-RL setting in the final version.

---

> ### Author Response · Authors · 2024-11-22
>
> **Q6: Lack of datasets.**
>
> Thanks for the suggestion! Our paper already included the results on **Ultrafeeback in Table 11** (page 22). Our choice of datasets aligns with recent prior work; for example, [11] uses the HH-RLHF dataset only. Also, we have demonstrated the effectiveness of CARDS on a set of RMs [7, 8, 9, 10]. These RMs are trained on a wider set of preference datasets.
>
> To further demonstrate CARDS on more datasets, we add the results on the suggested datasets [12, 13]. CARDS consistently outperforms previous work.
>
> | **Dataset** | **Method** | **RM Score** | **# LLM Calls** | **# RM Calls** | **# Total Calls** | **Inference Time per 100 samples (min)** |
> |:---:|:---:|:---:|:---:|:---:|:---:|:---:|
> | BeaverTails | ARGS | 7.93 | 128.00 | 5120.00 | 5248.00 | 126.3 |
> | BeaverTails | CARDS | 8.18 | 847.88 | 47.48 | 895.36 | 53.4 |
> | HelpSteer | ARGS | 6.55 | 128.00 | 5120.00 | 5248.00 | 818.38 |
> | HelpSteer | CARDS | 7.51 | 1046.76 | 73.80 | 1120.56 | 281.3 |
>
> [7] argsearch/llama-7b-rm-float32
>
> [8] weqweasdas/RM-Mistral-7B
>
> [9] Ray2333/reward-model-Mistral-7B-instruct-Unified-Feedback
>
> [10] weqweasdas/hh_rlhf_rm_open_llama_3b
>
> [11] ARGS: Alignment as Reward-Guided Search. ICLR 2024.
>
> [12] PKU-Alignment/BeaverTails
>
> [13] nvidia/HelpSteer
>
> **Q7: Eq. 4 is not well defined.**
>
> In Eq. 4, $s_{\geq t}$ is the sequence after the $t$-th token. It is sampled from the policy of base LLM $\pi_{LLM}(\cdot | s_{<t})$ given the previously generated prefix. We will modify the notation to $s_{\geq t}\sim \pi_{LLM}(\cdot | s_{<t})$ in the final version for clarity.
>
> **Q8: Which reward model did you use for Fig. 2b, 3, 4?**
>
> In these figures, both [7, 8] are used. Fig 3 shows reward distributions of [7, 8]. The points in Fig. 2b and Fig. 4 are collected from both [7, 8], demonstrating the relationship between prefix reward and full-sequence reward holds for diverse RMs.
>
> [7] argsearch/llama-7b-rm-float32
>
> [8] weqweasdas/RM-Mistral-7B
>
> **Q9: Can this approach be applied to some smaller and weaker reward models?**
>
> Yes. Aside from the standard results (7B base model, 7B RM) shown in the main body, we also have results for smaller RM (7B base model, 3B RM) in **Appendix B.7 (page 17)**. CARDS can still achieve promising alignment ratings and efficiency under this setting.
>
> **Q10: Can this approach be applied to multi-objective alignment?**
>
> Yes. As discussed in L43~L45, CARDS can be applied to the setting of multiple RMs, by aggregating the reward scores from these RMs. If each of these RMs represents a distinct objective, it is the case of multi-objective alignment. We believe this is an interesting direction and plan to explore this extension in future work.
>
> Thank you for your thorough feedback, which has greatly helped improve our work. We have carefully addressed the concerns raised and are happy to provide further clarification if needed.

---

> > ### Comment · Reviewer_udDg · 2024-11-22
> >
> > Thank you for your efforts. I will read them in details and respond to you later.

---

> ### Comment · Reviewer_udDg · 2024-11-22
> **raise my rating**
>
> Q1. I accept the clarification and I believe the claim would still make sense in many cases. As for the paper “from r to Q*”, I think the result can be directly extended to RM as long as the RM is trained to align BT-model, since you can just remove the $\pi_\text{ref}$ term in the proof.
>
> Q2. I have no relationship with this paper, and thus am not certain about the underlying reason. However, I think it is fine for different claims to be presented under different settings, for nlp community. I would suggest that the authors clarify the limitations of their claims and emphasize their empirical successes in the final revision.
>
> Q4. I understand that $V_\theta$ needs additional training. But I want to clarify that its inference cost is not large. Suppose you have a language model, you can directly get $\pi(s\vert x)$ for $\forall s\in\Omega$ at one time, since they are all matrix computation And the computation of $V_\theta(s\vert x)$ is the same as $\pi(s\vert x)$, since they may share the same architecture. Anyway, since the rebuttal phase draws to end, the existing experiments are already acceptable.
>
> Q6. Thank you for your efforts. I appreciate the results.
>
> I am satisfied about the explanation, and I will raise my rating to 6. But I strongly suggest the authors revising their paper:
>
> - remove the claims involving rigorous analysis
> - highlight the limitations of their claims as clarified in the rebuttal
> - emphasize the empirical study as the main contribution
>
> If the revision is not satisfying, I would reserve the right to decrease the rating.

---

> > ### Author Response · Authors · 2024-11-22
> >
> > Thank you for acknowledging our rebuttal and raising your score! We appreciate your engagement in the discussion. Based on your questions and our responses, we have updated the PDF with the following points (the added parts are marked in **blue**):
> >
> > - **Remove the claim of “rigorous” analysis and emphasize the empirical study:** L82, L90-L93, L209-L210, L220-L223, L244, L529.
> >
> > - **Highlight the limitations:** (1) We highlight that in Fig.2, the relation between prefix reward and full-response reward is based on 1 dataset and 2 RMs, and the results are dependent on appropriate choices of uncertainty threshold (page 5). (2) We highlight that the claim that “RM is a good approximation to the value function on semantically complete prefixes” is not a mathematical claim, but an observation based on empirical findings (page 5). (3) We highlight that Eq. 11b uses a property of value functions in the soft-RL setting (page 15).
> >
> > - **Additional experimental results:** We add the results on BeaverTails and HelpSteer (Appendix C.9 and Table 12).
> >
> > If you have further questions or concerns about the updated PDF, please let us know. We are happy to further revise the paper as needed. Thank you again for your valuable feedback.

---

### Official Review · Reviewer_QQnj · 2024-10-25

**Soundness:** 2
**Presentation:** 2
**Contribution:** 2
**Rating:** 5
**Confidence:** 4

**Summary:**

The paper introduces a decoding-time alignment method for large language models (LLMs) that ensures high-reward and high-likelihood text generation with reduced computational costs. CARDS uses rejection sampling to create text by dynamically determining segment lengths based on the predictive uncertainty of LLMs, significantly enhancing efficiency. This method also maintains fluency and aligns closely with human preferences, demonstrating substantial improvements over existing methods in both speed and alignment accuracy during text generation.

**Strengths:**

1. The article raises an excellent question on how to enhance the efficiency of alignment during the reasoning phase.
2. Many designs in the article's methodology are interesting, such as "Our method leverages the comprehension ability of pre-trained LLMs for segmentation."
3. The experimental results of CARDS are outstanding.

**Weaknesses:**

1. The presentation of the article is somewhat difficult to follow. For example, Figure 1, which explicates the contributions mentioned in the introduction, requires the integration of content from many sections later in the text to be understood. Moreover, Section 4.1.1 repeatedly refers to Figure 2c without clearly explaining how Figure 2c is produced and its detailed meaning. Section 4.2.1, however, does not mention Figure 2c at all.
2. Many intermediate conclusions in the methodology lack theoretical support and are merely based on simple experiments and conjectures. For instance, "a full-response reward will be high given a high-reward prefix" and "This is because initiating a new segment is more unpredictable than continuing an existing one."
3. I think previous works [1] also assume Lemma 1 is valid; however, this paper still does not provide convincing proof, thus, the contribution in this part seems incremental.
4. CARDS appears to be a method for enhancing the efficiency of decoding-time alignment, but I have no idea why CARDS could lead to significant performance improvements in the experiment.

[1] Args: Alignment as reward-guided search.

**Questions:**

1. What do you mean by "However, while some of the existing decoding-time alignment methods still struggle with the trade-off between alignment and fluency, they all encounter significant efficiency challenges due to auxiliary steps added to their decoding process."?
2. In line 239, "Therefore, the above observation also suggests that RMs can be used as value functions on semantically complete preffxes." Can you explain why?
3. In Lemma 1, I still cannot understand why can we assume "Assuming the reward models are equivalent to value functions when evaluating semantically complete prefixes". Is there any previous literature proof of this?
4. What if the prefixes is not semantically complete?

---

> ### Author Response · Authors · 2024-11-22
>
> **Q1: Fig. 1 requires integration from many sections later.**
>
> Fig. 1 is a high-level overview of our method, and thus it cannot explicitly show all of the technical details. There are **many new designs and findings** within our method and it is impossible to put them all together in one figure. We believe Figure 1 effectively conveys the core idea of our approach and the caption explicitly references the corresponding sections that elaborate on specific details. We would greatly appreciate your suggestions for improving Fig. 1 to make it clearer or more informative.
>
> **Q2: Section 4.1.1 repeatedly refers to Fig. 2c, while section 4.2.1 does not mention Fig. 2c.**
>
> Fig. 2c is to demonstrate that semantically complete segmentation is better than static segmentation in preserving the accuracy of RM scoring. Section 4.1.1 discusses the importance of semantically complete segmentation, which is closely related to Fig. 2c. Section 4.2.1 discusses the technical details of implementing uncertainty-based segmentation, which is a different topic from Fig. 2c. We believe referring to Fig. 2c only in section 4.1.1 is a reasonable structure.
>
> **Q3: Intermediate conclusions lack theoretical support.**
>
> This paper is primarily **empirical-result-driven**. We provide comprehensive **explanations and demonstrations** for the empirical findings in this paper. We believe empirical-result-driven insights and conclusions are also valuable. We are open to discussing the potential theoretical proofs for these explanations, but such mathematical work is beyond the scope of this paper.
>
> **Q4: [1] also assumes Lemma 1 is valid; this paper still does not provide convincing proof.**
>
> [1] has a stronger assumption than our Lemma 1. [1] assumes that the reward models are accurate on arbitrary prefixes, whereas our Lemma 1 only requires accuracy on semantically complete prefixes. This is grounded in the observation that RMs are typically trained on complete responses, and semantically complete prefixes closely resemble the data that RMs have seen during training. The improved alignment ratings and efficiency compared with [1] also support this claim.
>
> The proof of our Lemma 1 is rigorous, and the assumption is validated through empirical results—neither of which was done in the previous work [1].
>
> [1] ARGS: Alignment as Reward-Guided Search. ICLR 2024.
>
> **Q5: Why CARDS leads to significant performance improvements?**
>
> The way we use reward models is appropriate **(only scoring semantically complete sequences)**, and thus the reward scores are accurate in our framework. Previous works that use reward models on arbitrary prefixes suffer from inaccurate reward scores.
>
> Fig. 2c shows a good example of the inaccuracy of arbitrary prefixes. Reward scores on fix-length segments (which are not semantically complete) are not accurate.
>
> **Q6: "However, while some of the existing decoding-time alignment methods still struggle with the trade-off between alignment and fluency, they all encounter significant efficiency challenges due to auxiliary steps added to their decoding process."**
>
> Thanks for pointing out the confusion! The trade-off between alignment and fluency has been discussed by many papers [2, 3, 4], and the “auxiliary steps” refer to reward model scoring. These two statements may not be strongly correlated, and we will separate them into 2 sentences in the final version.
>
> [2] One fish, two fish, but not the whole sea: Alignment reduces language models' conceptual diversity. 2024.
>
> [3] A Roadmap to Pluralistic Alignment. 2024.
>
> [4] Does Alignment Tuning Really Break LLMs' Internal Confidence? BlackboxNLP 2024.
>
> **Q7: "Therefore, the above observation also suggests that RMs can be used as value functions on semantically complete prefixes."Can you explain why?**
>
> The mentioned observation shows that RMs are also accurate on semantically complete prefixes. This shares the same property as the value function in the discussed cases. Therefore, we conclude that RMs are approximately equivalent to the value function specifically on the semantically complete prefixes. This is **not** a mathematical claim, but an **explanation/analogy** based on empirical findings, as explained in our responses to Q3 and Q4.
>
> **Q7: "In Lemma 1, I still cannot understand why can we assume "Assuming the reward models are equivalent to value functions when evaluating semantically complete prefixes". Is there any previous literature proof of this?"**
>
> As mentioned in our response to Q4, we verify the assumption through empirical results. There is no previous literature proof of this and we are the first to empirically validate it.

---

> > ### Comment · Reviewer_QQnj · 2024-11-22
> > **Further Concerns**
> >
> > Thanks for your response. However, there are still some remaining concerns.
> >
> > About Q3, Q4 and the first Q7.
> >
> > I still think empirical results are insufficient for proofing a Lemma. I understand that empirical-result-driven insights are also valuable. However, the insights should be your experimental results, rather than methodology.
> >
> > About Q5.
> >
> > I think previous works like DPO and PPO also only scoring semantically complete sequences. DPO even does not use reward models.

---

> > > ### Author Response · Authors · 2024-11-22
> > >
> > > Thank you for your reply, allowing us the opportunity to clarify.
> > >
> > > **“I still think empirical results are insufficient for proofing a Lemma…However, the insights should be your experimental results, rather than methodology.”**
> > >
> > > Many impactful ML/DL papers introduce new methodologies without including any Lemmas. In fact, it is common in ML/DL research for methodologies and algorithms to be developed first, with theoretical proofs following later. In our work, we have provided clear motivation, explicitly stated the assumptions underlying our algorithm, and offered comprehensive empirical evidence to validate them. **The absence of a Lemma alone should not be the reason for rejection.**
> > >
> > > We believe that the strength of our contributions lies in the combination of a practical algorithm and rigorous empirical validation. Dismissing the work solely for lacking a Lemma overlooks the significant impact and insights it offers to the community. Instead, this paper provides a strong result for future theoretical developments while making immediate contributions to advancing methodology and understanding in the field.
> > >
> > > **”I think previous works like DPO and PPO also only scoring semantically complete sequences. DPO even does not use reward models.”**
> > >
> > > Thank you for your clarification. Compared to DPO/PPO, the improvement of CARDS comes from directly sampling from the target distribution without requiring training. This avoids issues like undesired convergence and difficult optimization during training, which have been noted in prior work [e.g. 1, 2]. The improvement of inference-time alignment methods over DPO/PPO has also been noted in [3].
> > >
> > > Compared to previous inference-time alignment methods such as RAD/ARGS, CARDS achieves better performance by only scoring semantically complete sequences.
> > >
> > > [1] Open problems and fundamental limitations of reinforcement learning from human feedback. TMLR 2023
> > >
> > > [2] Scaling Laws for Reward Model Overoptimization in Direct Alignment Algorithms. NeurIPS 2024
> > >
> > > [3] ARGS: Alignment as Reward-Guided Search. ICLR 2024.
> > >
> > > We sincerely appreciate the time you took to review our paper and read our responses. We were wondering if there are any specific flaws in the paper that led to your score of 3. Are there any changes we could make to address your concerns and potentially improve the score? Your input would be invaluable in helping us strengthen our work.

---

> > > > ### Comment · Reviewer_QQnj · 2024-11-23
> > > >
> > > > Thanks for your thoughtful response.
> > > >
> > > > For the first point,
> > > >
> > > > Please direct me to any 'impactful ML/DL papers' that introduce a new lemma without any proof or citation, relying solely on empirical evidence, as presented in your paper. If such examples are provided, I am willing to acknowledge the gaps in my understanding and increase my score.
> > > >
> > > > For the second point, sorry about that my previous question is somehow midleading. As far as I am aware, decoding-time alignment is not universally acknowledged to achieve significantly better alignment results compared to DPO or PPO. Therefore, I find this to be an impressive discovery and am curious about the reasons behind its success. Regardless, while I still have some reservations, this can be explored as part of future works.

---

> > > > > ### Author Response · Authors · 2024-11-23
> > > > >
> > > > > Thanks a lot for your reply!
> > > > >
> > > > > For the first point, we did not say that “impactful ML/DL papers that introduce a new lemma without any proof or citation, relying solely on empirical evidence”. What we said is that “Many impactful ML/DL papers introduce new methodologies without including any Lemmas”.  Regarding the claims in the methodology section, such as "a full-response reward will be high given a high-reward prefix" and "This is because initiating a new segment is more unpredictable than continuing an existing one", we did not present these as Lemmas in the paper. We explained their intuitions and provided comprehensive empirical support. **To avoid confusion, we will emphasize in the revised version that these claims are our explanations and observations, not mathematical claims.** Thank you for this feedback, which has greatly improved the clarity of the paper.
> > > > >
> > > > > In case you are referring to our Lemma 1, we have provided rigorous proof in Appendix A.
> > > > >
> > > > > For the second point, thank you for your clarification. We agree that further exploring the comparison between decoding-time alignment methods and fine-tuning-based approaches like DPO or PPO, as well as how they might complement each other, would be very interesting. We look forward to pursuing this in future work.
> > > > >
> > > > > Please let us know if you have any further questions; we would be happy to answer and discuss. Thank you again for engaging in this discussion!

---

> > > > > > ### Comment · Reviewer_QQnj · 2024-11-25
> > > > > >
> > > > > > Thanks for your response. I have raised my score.

---

> ### Author Response · Authors · 2024-11-22
>
> **Q8: What if prefixes are not semantically complete?**
>
> The case has been studied in Fig. 2c (page 5). The “Static” results are based on prefixes that are not semantically complete. RMs are no longer accurate when scoring such prefixes. This result verifies our assumption and highlights the effectiveness of our algorithm.
>
> Thank you for your constructive feedback, which has greatly helped strengthen our work. We have carefully addressed the concerns raised and are happy to provide further clarification if there are any additional questions.

---

### Official Review · Reviewer_KZCH · 2024-11-02

**Soundness:** 4
**Presentation:** 3
**Contribution:** 3
**Rating:** 6
**Confidence:** 3

**Summary:**

This paper propose a novel segment-based sampling method for efficient decoding-time alignment, leveraging rejection sampling to iteratively generate small semantic segments of high reward.

**Strengths:**

1.	This paper conducts a rigorous analysis of reward models and demonstrates the RMs can serve as value functions on semantically complete segments.
2.	The generation method is segment-based and the length of segment is dynamic.
3.	The experiment is adequate and reasonable and the paper is well written.

**Weaknesses:**

1.	The experiments are conducted on 7B models. The method could be verified on more larger models.
2.	The parallelization scheme of dynamic segmentation whether has a slow inference time when the batch size is larger.

**Questions:**

The same as above

---

> ### Author Response · Authors · 2024-11-22
>
> **Q1: Larger models.**
>
> The choice of model sizes follows recent prior work [1], where the largest model is 7B. To evaluate the performance of our method on larger models, we show below that CARDS continues to outperform ARGS on 13B models. The models used are from [2, 3]. Due to the limitation of GPU memory, these are the largest models we can run.
>
> | **Method** | **RM Score** | **# LLM Calls** | **# RM Calls** | **# Total Calls** | **Inference Time per 100 samples (min)** |
> |:---:|:---:|:---:|:---:|:---:|:---:|
> | ARGS | 2.67 | 128.00 | 5120.00 | 5248.00 | 275.8 |
> | CARDS | 3.24 | 441.80 | 7.72 | 449.52 | 74.2 |
>
> [1] ARGS: Alignment as Reward-Guided Search. ICLR 2024.
>
> [2] meta-llama/Llama-2-13b-chat-hf
>
> [3] miulab/llama2-7b-ultrafeedback-rm
>
> **Q2: Slow inference when batch size is large.**
>
> The overall inference speed is not compromised for larger batches. According to **Table 7 in page 17**, batch size 4 has fewer RM calls and its number of LLM calls is below 4 times of batch size 1. Importantly, for larger batches, LLM calls and RM calls are conducted parallelly. Therefore, the overall inference speed for larger batches is essentially faster.
>
> Thank you for your supportive and constructive feedback. We have carefully addressed the concerns raised, adding new results to strengthen our work. If there are any additional questions, we are happy to provide further clarification.

---

> ### Author Response · Authors · 2024-11-25
>
> Thank you for the valuable feedback! We have posted responses, and hope that they can address your concerns. We appreciate your time for the discussion and are happy to provide further explanations and/or experiments if you still have remaining questions.

---

### Official Review · Reviewer_VAa5 · 2024-11-04

**Soundness:** 3
**Presentation:** 3
**Contribution:** 3
**Rating:** 6
**Confidence:** 3

**Summary:**

- The paper focuses on enhancing decoding-time alignment methods, particularly addressing the high computational costs associated with existing approaches.
- It introduces **Cascade Reward Sampling (CARDS)**, a novel method that samples segments, leveraging the insight that reward models (RMs) favor high-reward prefixes. CARDS incrementally generates high-reward semantic segments, aiming to achieve optimal reward in the final output by prioritizing these segments.
- Experimental results demonstrate significant improvements in both performance and speed over baseline methods.

**Strengths:**

- The shift to segment-level sampling, along with the use of uncertainty as a termination signal for segments, presents a unique and novel approach.
- The results show significant gains on performance and speed, renders this approach very practical.

**Weaknesses:**

- The reliance on target score poses a notable limitation. How can one determine this? Different RMs provide values in different scales.
    - Exploring alternative sampling strategies, like sampling multiple segments per step and selecting the highest-scoring option before proceeding greedily, could be beneficial (atleast as a baseline comparison)
- The experiments are conducted solely on the HH-RLHF dataset, limiting the generalizability of the findings. Especially, HH-RLHF is a very simple dataset as many inference time methods are already better than trained methods such as PPO and DPO. Testing on few more datasets would solidify the findings.

**Questions:**

Suggestions:
- One could include more baselines such as In-Context learning for alignment (Rethinking alignment via in-context learning), Best of N based on full sequence rewards

---

> ### Author Response · Authors · 2024-11-22
>
> **Q1: The reliance on target scores.**
>
> The reliance on target reward score $r^\star$ is a general feature of rejection-sampling-based methods like [1]. In practice, $r^\star$ is estimated by a small subset of chosen responses in the preference dataset, and the **same strategy** is used in [1].
>
> Alternative strategies like parallelly sampling multiple candidates and choosing the best one (segment-level best-of-$N$ searching) do not depend on $r^\star$. However, they require specifying the value of $N$ and face significant efficiency issues, as the number of sampled candidates remains fixed at every step.
>
> We have implemented the segment-level best-of-$N$ searching to compare the efficiency. We adopt the same uncertainty-based segmentation as used in CARDS. $N$ is set to 10, and the experiment is based on Llama 7B and HH-RLHF. The results below show that segment BoN improves over ARGS, which is token-level BoN, but is much slower than CARDS.
>
> | **Method** | **RM Score** | **# LLM Calls** | **# RM Calls** | **# Total Calls** | **Inference Time per 100 samples (min)** |
> |:---:|:---:|:---:|:---:|:---:|:---:|
> | Token BoN (ARGS) | 7.85 | 128.00 | 5120.00 | 5248.00 | 238.7 |
> | Segment BoN | 8.47 | 1846.72 | 68.18 | 1914.90 | 195.9 |
> | CARDS | 8.71 | 744.14 | 34.48 | 778.62 | 66.1 |
>
> [1] Statistical Rejection Sampling Improves Preference Optimization. ICLR 2024.
>
> [2] ARGS: Alignment as Reward-Guided Search. ICLR 2024.
>
> **Q2: The experiments are conducted solely on the HH-RLHF dataset.**
>
> We do have results of the UltraFeedback dataset in **Table 11 (page 22)**, where the alignment rating and efficiency are consistent with the HH-RLHF results. Our choice of datasets aligns with recent prior work; for example, [2] uses the HH-RLHF dataset only.
>
> To further demonstrate the generalizability of our findings, we also provide the results on two more datasets [4, 5]. CARDS consistently outperforms previous work.
>
> | **Dataset** | **Method** | **RM Score** | **# LLM Calls** | **# RM Calls** | **# Total Calls** | **Inference Time per 100 samples (min)** |
> |:---:|:---:|:---:|:---:|:---:|:---:|:---:|
> | BeaverTails | ARGS | 7.93 | 128.00 | 5120.00 | 5248.00 | 126.3 |
> | BeaverTails | CARDS | 8.18 | 847.88 | 47.48 | 895.36 | 53.4 |
> | HelpSteer | ARGS | 6.55 | 128.00 | 5120.00 | 5248.00 | 818.38 |
> | HelpSteer | CARDS | 7.51 | 1046.76 | 73.80 | 1120.56 | 281.3 |
>
> [2] ARGS: Alignment as Reward-Guided Search. ICLR 2024.
>
> [4] PKU-Alignment/BeaverTails
>
> [5] nvidia/HelpSteer
>
> **Q3: More baselines such as in-context learning for alignment and best-of-$N$ on full sequences.**
>
> We primarily choose baselines following recent previous works like ARGS [2] and RAIN [3]. Notably, we have already included RAIN [3], an in-context learning method, as our baseline.
>
> Previous works also exclude BoN as a baseline due to its high computational cost. For completeness, we now provide results for BoN below. Full-sequence BoN achieves worse alignment ratings and is slower than CARDS.
>
> | **Method** | **RM Score** | **# LLM Calls** | **# RM Calls** | **# Total Calls** | **Inference Time per 100 samples (min)** |
> |:---:|:---:|:---:|:---:|:---:|:---:|
> | BoN | 7.64 | 1280.00 | 10.00 | 1290.00 | 116.0 |
> | CARDS | 8.71 | 744.14 | 34.48 | 778.62 | 66.1 |
>
> [2] ARGS: Alignment as Reward-Guided Search. ICLR 2024.
>
> [3] RAIN: Your Language Models Can Align Themselves without Finetuning. ICLR 2024.
>
> Thank you for your supportive and constructive feedback. We have carefully addressed the concerns raised, adding new results and baselines to strengthen our work. If there are any additional questions, we are happy to provide further clarification.

---

> ### Author Response · Authors · 2024-11-25
>
> Thank you for the valuable feedback! We have posted responses, and hope that they can address your concerns. We appreciate your time for the discussion and are happy to provide further explanations and/or experiments if you still have remaining questions.

---

### Author Response · Authors · 2024-11-22
**Summary Response to All Reviewers and AC**

We thank all reviewers for their detailed and constructive reviews. The key concern from the reviewers is the conclusion that **RMs are approximately value functions for semantically complete prefixes**. We would like to emphasize that this conclusion is based on **empirical results**, supported by comprehensive **explanations and demonstrations** provided in the paper. We believe empirical-result-driven insights and conclusions are also valuable.

We have made the following changes to address reviewers’ concerns:

- We provided results on new datasets: BeaverTail and HelpSteer.

- We provided results for an additional baseline: full-sequence best-of-$N$ searching.

- We provided results on larger models: Llama2 13B

This paper offers novel insights into the generalization of RMs on incomplete responses, introduces a simple and effective alignment algorithm, and demonstrates significant empirical improvements. We believe this paper will be valuable to research on reward models and alignment.

---

### Meta-Review · Area_Chair_DBZM · 2024-12-22

**Metareview:**

The paper proposes a new technique for reward guided text generation that is computationally cheaper than previous methods and achieves higher scores.  The strength of this paper is its empirical evaluation that demonstrates clear improvements over the baselines.  The weakness of this paper is the absence of theory, a problematic assumption and questionable analysis.  I recommend rejection since the analysis has important inconsistencies.

**Additional Comments On Reviewer Discussion:**

The reviewers see some good potential in the proposed approach.  The empirical results are good and demonstrate clear improvements over the baselines.  The problem is the questionable analysis and problematic justification provided to explain the approach and the results.  In addition to the comments already provided in the reviews, let me articulate some problems and inconsistencies.

First Eq 4 is problematic.  Eq 4 suggests that the value of a prefix is the expected value of the completion of this prefix according to the LLM distribution.  This is problematic since the value of a prefix should reflect human preferences only instead of depending on the LLM.  When the values of prefixes depend on the LLM, then if we change the LLM that we are trying to improve then the value of those prefixes will also change.  The point of RLHF is to use preference data to improve an LLM, but if we use this LLM to define the values of prefixes then we are effectively using the LLM that is assumed to be inaccurate to improve itself based on its own inaccurate distribution.

Second, as noted by the reviewers the assumption in Lemma 1 is problematic.  Lemma 1 assumes that $r(x,y_{<t}) = V(x,y_{<t})$.  When we combine this assumption with Eq 4 we get $r(x,y_{<t}) = \sum_{y_{>t}} \pi(y_{>t}|x,y_{<t}) V(x,y)$.  Again, this means that the reward of a prefix is dependent on the LLM probabilities, which is problematic.

Third, in Figure 2a, the paper shows the empirical value distribution of full sequences given a prefix and suggests that this distribution is approximately Gaussian with its mean controlled by the value of the prefix as shown in Eq 5. This is inconsistent with Eq 4 since Fig 2a and Eq 5 rely on the preference data distribution to estimate this Gaussian while Eq 4 relies on the LLM distribution.

Overall, it is not clear why the proposed approach should achieve higher scores than the baselines as demonstrated in the empirical results.

---

### Decision · Program_Chairs · 2025-01-22

Reject